# Associations between Pleurisy and the Main Bacterial Pathogens of the Porcine Respiratory Diseases Complex (PRDC)

**DOI:** 10.3390/ani13091493

**Published:** 2023-04-27

**Authors:** Fernando Antônio Moreira Petri, Geovana Coelho Ferreira, Laíza Pinto Arruda, Clarisse Sena Malcher, Gabriel Yuri Storino, Henrique Meiroz de Souza Almeida, Karina Sonalio, Daniela Gomes da Silva, Luís Guilherme de Oliveira

**Affiliations:** 1School of Agricultural and Veterinarian Sciences, São Paulo State University (Unesp), Jaboticabal 14884-900, SP, Brazil; 2Unit of Porcine Health Management, Faculty of Veterinary Medicine, Ghent University, Salisburylaan 133, 9820 Merelbeke, Belgium

**Keywords:** pig production, slaughterhouse, SPES

## Abstract

**Simple Summary:**

Respiratory disease is a significant health issue in intensive pig production, and it can have a significant impact on animal welfare, productivity, and profitability. Some of the common respiratory diseases affecting pigs include Porcine Respiratory Diseases Complex (PRDC). These pathogens can act synergistically and cause severe respiratory disease in pigs. Hence, to document and quantify the occurrence of these pathogens, 1015 carcasses of pigs were evaluated according to the pleuritis score in a slaughterhouse. Lungs were evaluated according to the pneumonia index (0.91 to 1.89) and pleurisy score were determined. Samples of lung and pleura were collected for DNA quantification by PCR of these five pathogens in fattening pigs. Moreover, the correlation of bacterial quantification and macroscopic lesions were performed. Our finding revealed a prevalence of 12.3% of pleuritis and correlations between quantification of *Actinobacillus pleuropneumoniae*, *Glaesserella parasuis*, *Mycoplasma hyopneumoniae*, *Pasteurella multocida*, and *Streptococcus suis* between different pleurisy scores. Finally, monitoring respiratory lesions in slaughter pigs can provide valuable information about the efficacy of control measures that have been implemented, allowing veterinarians and pig producers to make informed decisions about future management strategies that can lead to improved diagnosis, treatment, and prevention of PRDC.

**Abstract:**

Porcine Respiratory Diseases Complex (PRDC) is a multifactorial disease that involves several bacterial pathogens, including *Mycoplasma hyopneumoniae (M. hyopneumoniae)*, *Actinobacillus pleuropneumoniae (A. pleuropneumoniae)*, *Pasteurella multocida (P. multocida)*, *Glaesserella parasuis (G. parasuis)*, and *Streptococcus suis (S. suis).* In pigs, the infection may cause lesions such pleurisy, which can lead to carcass condemnation. Hence, 1015 carcasses were selected from three different commercial pig farms, where the respiratory conditions were evaluated using slaughterhouse pleurisy evaluation system (SPES) and classified into five groups. In total, 106 pleural and lung fragments were collected for qPCR testing to identify the five abovementioned pathogens. A moderate correlation between the severity of the lesions and the presence of *P. multocida* (R = 0.38) and *A. pleuropneumoniae* (R = 0.28) was observed. Concerning the lung samples, the severity of the lesions was moderately correlated with the presence of *P. multocida* (R = 0.43) and *M. hyopneumoniae* (R = 0.35). Moreover, there was a strong correlation between the presence of *P. multocida* and *M.hyopneumoniae* in the pleura (R = 0.82). Finally, this approach may be a useful tool to identify and quantify causative agents of PRDC using qPCR, providing a comprehensive evaluation of its relevance, strength, and potential application in the field as a surveillance tool for veterinarians.

## 1. Introduction

Respiratory diseases are a leading cause of economic losses in the swine industry around the world, attributable to increased mortality rates, decreased average daily weight gain (ADWG), condemnation of carcasses in slaughterhouses, and the expenses incurred for their control and prevention. Environmental conditions, management factors, population size, and factors such as age and genetics all play important roles in the development of Porcine Respiratory Diseases Complex (PRDC) [1,2]. The PRDC is caused by a range of pathogens including porcine circovirus type 2 (PCV-2), porcine cytomegalovirus, porcine reproductive and respiratory syndrome viruses 1 and 2 (PRRSV-1, 2), swine influenza A virus (swIAV), *Actinobacillus pleuropneumoniae (A. pleuropneumoniae)*, *Bordetella bronchiseptica (B. bronchiseptica)*, *Glaesserella parasuis (G. parasuis)*, *Mycoplasma hyopneumoniae (M. hyopneumoniae)*, *Mycoplasma hyorhinis (M. hyorhinis)*, *Pasteurella multocida (P. multocida),* and *Streptococcus suis (S. suis)*. While PRDC is extensively studied [3], there is a lack of information on the role of coinfections in the development of pleuritis, which is the inflammation of the pleura, a common respiratory disease in pigs. The PRDC is most commonly observed in fattening and finishing pigs, with mortality rates ranging from 2 to 10% and morbidity rates ranging from 10 to 40%. Frequent clinical signs of PRDC include coughing, fever, dyspnea, decreased feed intake, and even fatal pneumonia. The association of lung lesions, such as pleurisy and lung scars, with poor performances in terms of decreased average daily weight gain (ADWG) and economic return [4,5], highlights the importance of monitoring lesions at the slaughterhouse. The most common lesions observed during post-mortem evaluations are pneumonia and pleurisy [6]. In the Brazilian pig industry, Galdeano et al. (2019) [7] and Baraldi et al. (2019) [8] reported a prevalence of pleurisy lesions of 9 and 14%, respectively.

Pleurisy is a condition characterized by inflammation of the visceral and parietal pleura, the membranes that cover the lungs and line the chest cavity, respectively. In chronic cases, fibrotic adhesions may form between the parietal and visceral membranes of the pleural sac [9,10]. Bacterial infections such as *A. pleuropneumoniae* [6,11,12,13], *M. hyopneumoniae* [6,8], and *P. multocida* [9,14] can cause pleurisy in pigs, as well as viral infections, environmental stressors, and poor management practices. These adhesions can be observed during post-mortem evaluations of pig carcasses in slaughterhouses [15]. These evaluations, which involve the visual inspection of organs and tissues to detect abnormalities and lesions, are mandatory in slaughterhouses and are relatively easy and practical to implement. The lungs, pleura, liver, and heart are the most commonly observed sites of chronic pleurisy and cranioventral pulmonary consolidation (CVPC) lesions [16,17], with pneumonia and pleurisy being the most prevalent conditions observed during these assessments. The evaluation of lesions in pig carcasses is important for identifying potential health issues and ensuring the safety and quality of pork products for consumers [11,16].

The health condition in herds can be dynamic, and one surveillance tool alone is not entirely independent of other surveys conducted by the veterinarian on the farm and slaughterhouse. Monitoring lung lesions in pigs is an essential tool for assessing risk factors on farms and implementing prevention or control measures [7,12]. This should be done whenever low performance, mortality, or the occurrence of clinical signs suggestive of respiratory diseases in farm animals are observed. The slaughterhouse plays a crucial role in collecting data to evaluate the health status of the herd and the occurrence and severity of lung injuries caused by respiratory pathogens [11,12]. Moreover, veterinarians can visually examine slaughtered pigs and assess the presence of pleurisy and CVPC lesions at the slaughter line. However, abattoir examinations can be challenging due to the difficulties in record-keeping and are not suitable for establishing an accurate etiological diagnosis. This type of monitoring is mostly performed if the veterinarian intends to determine the success of vaccination and health programs on the farms [18]. During the post-mortem assessment, the respiratory organs of pigs are meticulously examined for PRDC signs, such as lung, trachea, heart, liver, and cavity chest lesions. The examination may also involve taking samples for laboratory analysis, including tissue fragments, nasal, tracheal, and lung swabs, which can be used to confirm the presence of pathogens and confirm the disease, in accordance with the clinical evaluation of animals carried out on the farm by the veterinarian.

Numerous scoring systems have been devised to assess respiratory lesions in pigs [19]. Among these, the slaughterhouse pleurisy evaluation system (SPES) [20] is extensively employed to quantify pleurisy, which is primarily caused by *A. pleuropneumoniae* and *M. hyopneumoniae* infection [12,13]. The SPES scores lesions on a scale of 0 to 4, based on the degree of involvement, and evaluates lung lesions by considering the weight of the affected lung lobe. The results are reported on a 100-point scale [17,21]. The pneumonia index (PI) methodology is a widely used approach by veterinarians and meat inspectors to categorize pneumonia lesions in pig carcasses at slaughterhouses. Madec and Kobisch developed this system in 1982 [22]. The PI assesses the percentage of each lung lobe affected and scores ranging from 0 to 4 based on the respective areas of pulmonary consolidation in lobes. Piffer and Brito adapted the system in 1991 [23], and it includes an index that summarizes the respiratory status of the herds. By employing these scoring systems, farmers and veterinarians can monitor the respiratory health of pigs and take measures to prevent and control respiratory diseases [24,25].

In a study conducted by Trevisan et al. [26] in Brazil, lungs with macroscopic alterations were evaluated after being diverted from the slaughter line to the Department of Final Inspection (DIF). The study found that 33.61% (41/122) of the lungs showed pleurisy, which was similar to the observation made by Valença et al. [27], who reported that 65.2% of the lungs presented lung parenchymal adhesions, making it difficult to accurately assign the percentage of the affected area. These findings emphasize the importance of accurately characterizing the location and extent of lung lesions in pigs for developing effective control and prevention measures.

By monitoring and sampling respiratory lesions at slaughter, veterinarians can identify the prevalence and severity of pleurisy lesions in a particular population of pigs, and implement targeted control measures, such as vaccination programs or improved herd management practices, to reduce the incidence and impact of pleurisy. Scoring systems, such as SPES and PI, aid in the accurate assessment of lung lesions and facilitate effective disease management strategies [19]. Specific tests such as serology, pathogen isolation, or molecular detection are necessary for diagnosis of the pathogens present in these lesions [28,29]. Among the molecular techniques, the quantitative polymerase chain reaction (qPCR) is noteworthy as it can identify several existing pathogens at once, making it a great alternative for the detection of the microorganisms involved in the PRDC [30,31,32]. Therefore, this study aimed to investigate the associations between the presence of bacterial pathogens in post-mortem pleurisy lesions classified by the SPES method in both pleura and lung fragments, as well as to evaluate how these pathogens interact with each other in relation to the occurrence of pleurisy in commercial pigs at slaughter.

## 2. Materials and Methods

### 2.1. Study Design, Lung and Carcass Evaluation, and Sample Collection

The study was carried out at a swine slaughterhouse located in Guariba, São Paulo State, Brazil. The slaughterhouse has a daily flow of approximately 400 animals from different regions of Brazil. Pigs between six and eight months of age and weighing between 100 and 130 kg were slaughtered. The research procedures were submitted for approval to the Ethics Committee on the Use of Animals (CEUA) of FCAV/Unesp-Campus Jaboticabal under the protocol number 21/002326.

The number of samples was determined using EpiInfo software (EpiInfo version 7.2.2.6-CDC, Atlanta, GA, USA), considering an error of 5% for a population of 120 animals per lot, an expected occurrence of 52% of pleuritis [33], and five groups. To estimate the occurrence of the five pathogens, a sample size of 10 per group (10 lung and 10 pleural samples) was obtained. To increase the representativeness of the study, collections were carried out on at least six different occasions, ensuring a representative number of samples per group.

A total of 1015 animals from three different commercial pig farms in Brazil were evaluated at the slaughterhouse on six different dates. The carcasses were selected randomly and based on the availability of the slaughterhouse. The pigs were slaughtered according to the methods established by ordinance No. 711, 1 November 1995, of the Ministry of Agriculture, Livestock and Food Supply [34]. After evisceration, the respiratory set (trachea and lung) of selected animals was identified and individually separated at the slaughter line.

The lungs of all pigs were evaluated for the presence and severity of macroscopic lesions. Each lobe was individually evaluated to estimate the percentage of the affected area, and a score from 0 to 4 was assigned based on the extent of the lesion, as described in the literature [23]. The score for each lobe was then multiplied by the relative lung weight to calculate the total area injured. As the lobes did not represent equal parts of the total lung volume, the following relative percentages were assigned: right apical lobe = 11%, right cardiac lobe = 11%, right diaphragmatic lobe = 34%, left apical lobe = 6%, left cardiac lobe = 6%, left diaphragmatic lobe = 27%, intermediate lobe = 5% [22,23]. The degree of injury was assessed according to the total area of pneumonia, using the mean of each lobe score concerning the total lung area to obtain the PI [23,35,36]. Groups with a PI mean of up to 0.55 were considered pneumonia-free (Score 0). Animals with a mean PI between 0.56 and 0.89 obtained an intermediate classification (Score 1) in which the presence of pneumonia occurred. Animals with a PI above 0.90 were considered severely affected, with high levels of pneumonia (Score 2). Carcasses and lungs that presented pleurisy lesions were classified using the SPES method as described by Dottori et al. (2007) [20], as described in Table 1.

During the routine processing of these carcasses, fragments of lungs and pleura were conveniently sampled and grouped based on the degree of pleurisy observed during macroscopic evaluation using SPES. The five groups were classified as follows: G1 included samples with a pleurisy score of 1; G2 included samples with a pleurisy score of 2; G3 included samples with a pleurisy score of 3; G4 included samples with a pleurisy score of 4; and G5 included samples with a pleurisy score of 0, which served as the control group. This grouping allows us to compare the molecular characteristics of lung and pleura samples across different levels of pleurisy severity and to determine any association between bacterial pathogens infection and pleurisy severity (Figure 1).

After identifying the carcass and determining the pleurisy score, small fragments of the parietal pleura adhered to the carcass, as well as representative fragments of the apical–cardiac–diaphragmatic lobes of its corresponding lung, were collected for qPCR. These lobes are commonly evaluated in swine respiratory health assessments. The lung set, comprising cranioventral and diaphragmatic lobe fragments, was collected in a representative manner based on observed lesions, with bronchi and lung parenchyma included in the aliquots.

All samples were collected with sterile instruments and scissors and deposited into sterile RNAse and DNAse-free microtubes (Eppendorf^®^, Hamburg, Germany). The samples were preserved in liquid nitrogen and transported to the Swine Medicine Laboratory at Unesp/FCAV Jaboticabal, where they transferred to a −20 °C freezer until further processing.

### 2.2. DNA Extraction and PCR for the Gadph Gene

An in-house extraction protocol was employed to extract total DNA, as previously described in the literature [37], using 0.04 g of tissue fragment. The concentration of DNA was determined by spectrophotometry using a NanoDrop™ One Spectrophotometer (Thermo Fisher Scientific^®^, Wilmington, DE, USA). To minimize the possibility of false negative results in the qPCR assay, all DNA samples were tested for the *gapdh* gene, using the primers gapdh-F (5′-CCTTCATTGACCTCAACTACAT-3′) and gapdh-R (5′-CCAAAGTTGTCATGGATGACC-3′), which flank a fragment of 437 base pairs (bp). The conventional PCR (cPCR) was performed as previously described by Birkenheuer et al. [38], with some modifications [39].

### 2.3. Multiplex qPCR for A. pleuropneumoniae, M. hyopneumoniae and P. multocida

To perform absolute quantification, qPCR was carried out on all DNA samples extracted from lung and pleura. The samples were tested in duplicate following the Minimum Information for Publication of Quantitative Real-Time PCR Experiments (MIQE) guidelines [40]. The multiplex qPCR targets were the *omlA* gene (virulence protein), the *p102* gene (adhesin), and the *kmt1* gene (membrane lipoprotein) from *A. pleuropneumoniae*, *M. hyopneumoniae,* and *P. multocida*, respectively. The primers and probes used for the qPCR were previously described and are listed in Table 2.

The qPCR reaction consisted of 2 μL of DNA template, 0.3 μM of each hydrolysis probe, 0.5 μM of each initiator oligonucleotide, 1X Master Mix GoTaq^®^ (Promega, Madison, WI, USA), and nuclease-free water (Promega^®^, Madison, WI, USA) to a total volume of 10 μL. Amplification was performed using a CFX96™ Real-Time PCR Detection System thermocycler (Bio-Rad, Hercules, CA, USA) with an initial 3 min denaturation cycle at 95 °C, followed by 39 cycles at 95 °C for 15 s and annealing/extension at 57.2 °C for 30 s. All samples were tested in duplicates, and the results were accepted only for those with a standard deviation lower than or equal to 0.5 cycles [40]. Samples with a deviation greater than 0.5 were retested in triplicates. The curves of the fluorophores FAM (target *p102*), Cy5 (target *omIA*), and TexRd (target *kmt1*) were analyzed, and the results were visualized with Bio-Rad CFX Manager Version 3.0 (Bio-Rad, Hercules, CA, USA). For absolute quantification of DNA in the samples, serial dilutions were prepared to determine standards with different concentrations of synthetic DNA (Gblock^®^, IDT, Iowa City, IA, USA) containing the target sequence (1 × 10^7^ copies/μL at 1.0 × 10^1^ copies/μL), for construction of the reaction standard curve. The qPCR assays followed the MIQE [40].

### 2.4. qPCR Assays for G. parasuis and S. suis

To detect *S. suis* serotype 2 and *G. parasuis,* a singleplex qPCR assay was performed for each pathogen. For *S. suis* detection, the *gdh* gene was targeted using oligonucleotide primers F (5′-GGTTACTTGCTACTTTTGATGGAAATT-3′) and R (5′-CGCACCTCTTTTATCTCTTCCAA-3′) [41]. For *G. parasuis* detection, the *infB* gene was targeted using oligonucleotide primers F (5′-CGACTTACTTGAAGCCATTCTTCTT-3′) and R (5′-CCGCTTGCCATACCCTCTT-3′) [43]. The qPCR reactions were performed using Quantitect^®^ SYBRGreen master mix (Qiagen, Germantown, MD, USA) and 1 μL of DNA template in a total reaction volume of 10 μL. qPCR was carried out on a CFX 96 thermocycler (Bio-Rad, Hercules, CA, USA). The specificity of the amplicons was assessed by a dissociation curve, and a variation of ±0.5 °C was accepted. Amplicon size and dissociation temperature for each reaction are shown in Appendix A. The efficiency of each reaction was determined by serial 10-fold dilutions (ranging from 10^7^ copies/μL to 10^1^ copies/μL) of positive controls (Gblock^®^, IDT, Iowa City, IA, USA) containing the amplified fragment of each primer pair. The reaction was only validated if the efficiency was between 90 and 105% [40].

### 2.5. Data Analysis

To assess data normality, we used the Shapiro–Wilk test, and Bartlett’s test was used to test for homogeneity of variances. For parametric data, we used a two-tailed ANOVA to compare the different groups, followed by the Tukey test for multiple comparisons. The Chi-squared test was used to further compare the scores. In the case of non-parametric data, we used the Kruskal–Wallis test to detect significant differences between groups, followed by the Dunn test for multiple comparisons. To detect correlations, we used Pearson’s correlation test for parametric data and Spearman’s rank correlation test for non-parametric data, both considering *p* < 0.05. Since multiple correlations were tested, we corrected the *p*-value using the Bonferroni correction to avoid errors. All calculations were performed using R software version 3.5.1 (R Core Team, 2018).

## 3. Results

### 3.1. Lungs and Carcass Evaluation

In this study, we sampled a subset of 106 carcasses out of a total of 1015 that were evaluated and scored from three different pig farms at the slaughterhouse. All evaluated herds showed some level of pleurisy. Based on the assessment of pulmonary consolidation and classification of pleuritis, the PI and percentage of pleurisy were calculated for each herd (as shown in Table 3). The PI values ranged from 0.91 to 1.89, while the percentage of carcasses with pleurisy in the herds ranged from 5.95% in lot 3 to 18.92% in lot 4. Furthermore, the percentage of lung consolidation varied among herds, with the lowest being 5.21% and the highest being 16.32%.

### 3.2. Multiplex qPCR Assays for A. pleuropneumoniae, M. hyopneumoniae, P. multocida, and qPCR Assays for G. parasuis and S. suis

Out of the 53 lung and 53 pleural samples (*N* = 106) that were submitted for the endogenous cPCR for *gapdh* gene, all were positive and subjected to multiplex qPCR assays for *A. pleuropneumoniae*, *M. hyopneumoniae*, and *P. multocida*, using the *omlA* gene, *p102* bacterial and *kmt1* gene, respectively, in duplicate. The results were analyzed based on three different fluorophores, and the average cycle quantification (Cq) and starting quantification (SQ) values of each sample were reported. The analysis revealed that 92.45% (49/53) of the lung samples and only 24.53% (13/53) of the pleura samples tested positive for *M. hyopneumoniae*. Additionally, 56.6% (30/53) of the lung samples and 39.6% (21/53) of the pleura samples tested positive for *A. pleuropneumoniae.* Moreover, 39.6% (21/53) of the lung samples were positive for *P. multocida,* and only 11.32% (6/53) of the pleura samples tested positive for *P. multocida*.

For the multiplex qPCR assays, all samples were run in duplicates on three different plates, with reaction efficiencies ranging from 93.5 to 102.8%. The analytical assay sensitivity was 10^1^ numbers of target genes copies/μL. The slope values ranged from −3.568 to −3.212, the determination coefficient (*R*^2^) values ranged from 0.991 to 0.999, and the y-Int ranged from 35.97 to 37.156 (Appendix A).

Of the 106 samples positive for the endogenous *gapdh* gene, all were tested in duplicates for *G. parasuis* and *S. suis*, using the *infB* gene and *gdh* gene, respectively. The average cycle quantification (Cq) and starting quantification (SQ) values were calculated, and the results showed that 90.57% (48/53) of the lung samples and 98.1% (52/53) of the pleura samples tested positive for *G. parasuis* in qPCR, whereas 92.45% (50/53) of the lung samples and 79.24% (42/53) of the pleura samples tested positive for *S. suis* in qPCR. Detailed results are presented in Table 4, and all qPCR assay parameters are listed in Appendix A.

The pleural samples that tested positive for *M. hyopneumoniae* showed no statistical difference between the pleuritis scores (*p* > 0.05), while in lung samples, score 0 differed significantly from score 4 (*p* = 0.02), while the other scores showed no significant difference. The pleural samples that tested positive for *P. multocida* showed a statistical difference (*p* = 0.01) between score 4 and scores 0, 1, and 2, while the other scores showed no significant differences. In lung samples, there was a difference between score 0 and score 4, and between scores 1 and 4. Regarding the *A. pleuropneumoniae* positive samples, there was statistical difference between the quantification of bacteria in lung samples and the pleurisy scores 0 and 2 and 3 and between scores 1 and 2, 3, and 4. There was no significant difference between the pleural and lung samples positive for *G. parasuis* and *S. suis*. Table 5 presents the detailed results for qPCR quantification of each pathogen according to the pleurisy score.

To evaluate lesions in the respiratory tracts during pathological investigations, we conducted a correlation analysis between qPCR values in lung and pleura samples and SPES scores. The analysis revealed a linear correlation between lung and pleural scores (*p* < 0.05). The test showed significant positive correlations between the DNA loads of *M. hyopneumoniae* and *P. multocida* in qPCR and the gross and macroscopic lung lesions and scores of pleuritis in carcasses (*p* = 0.03). However, the correlation was moderate (Spearman’s R coefficient < 0.5) due to the wide distribution of pathogens quantification in each class of SPES scores. Lung scores were divided into three categories, while SPES scores were divided into five categories. The results showed a high frequency of severe pleural lesions corresponding to a high frequency of severe lung lesions. However, there were no significant correlations for *G. parasuis* and *S. suis* with the same variables.

In the coinfection scenario, we found a strong correlation between the presence of *P. multocida* and *M. hyopneumoniae* in samples of pleura (*p* = 0.01, R = 0.82). In the lung samples, we observed a negative correlation between the presence of *A. pleuropneumoniae* and *M. hyopneumoniae* (*p* = 0.04, R = −0.42) and a positive correlation between the presence of *A. pleuropneumoniae* and *S. suis* (*p* = 0.04, R = 0.45) at pleuritis score 2. We also found a positive correlation between the presence of *P. multocida* and *M. hyopneumoniae* (*p* = 0.03, R = 0.58) and *M. hyopneumoniae* and *S. suis* (*p* = 0.02, R = 0.63), at score 4. Finally, there was no correlation between the presence of *S. suis* and *G. parasuis* between the groups of pleurisy lesions (Table 6).

## 4. Discussion

Bacterial pathogens, such as *M. hyopneumoniae*, *P. multocida*, *A. pleuropneumoniae*, *B. bronchiseptica*, *G. parasuis*, and *S. suis* are commonly associated with PRDC, which can result in chronic pulmonary lesions in pigs. Detecting these pathogens may require sampling of lung tissue and parietal pleura, as the infection can spread from the lungs to the chest cavity. In this study, macroscopic lung lesions and parietal pleura samples from slaughtered pigs were used to assess the etiology of pleuritis and, in particular, the role of *P. multocida* and *M. hyopneumoniae*, known causes of porcine pleuropneumonia and predispose pigs to secondary infections. Jirawattanapong et al. (2010) [44] reported the involvement of *P. multocida* and *A. pleuropneumoniae* in pleuritis lesions in 8 out of 10 pig farms [44]. Enøe et al. [45] conducted research and found that herds seropositive for *M. hyopneumoniae* or *A. pleuropneumoniae* had a higher prevalence of chronic pleurisy and CVPC in pigs at the time of slaughter. Specifically, the study found that 29% of pigs in herds seropositive for *M. hyopneumoniae* had CVPC, and 51% of pigs in herds seropositive for *A. pleuropneumoniae* had chronic pleurisy. 

Recent studies have shown that the prevalence of animals with pulmonary consolidation at slaughter in Brazil is alarming, and a similar prevalence rate has been found in different regions of the country. Baraldi et al. [8] evaluated 21 commercial farms in the State of São Paulo in 2016 and 2017, and found pleuritis and CVPC in 72.4% of evaluated carcasses, with an average consolidated lung area of 12%. Another study evaluated 30 herds of commercial pigs from the state of Goiás, which had a prevalence rate of 10.3 and 80.3% of pleuritis and CVPC, respectively [7]. In the state of Rio Grande do Sul, *M. hyopneumoniae* was detected in 83.3% of the samples analyzed for different pneumonia-causing agents in pigs slaughtered from five integrating companies [46]. In the State of Minas Gerais, 68.5% (333/486) of the lungs evaluated in a fattening herd showed CVPC and pleurisy [47].

In our study, the PI values ranged from 0.91 to 1.89, indicating that all herds were classified as having moderate to severe pneumonia conditions. Herds with a PI value between 0.51 and 0.99 are considered moderate, while a value more than 1.0 is considered severe, indicating that pneumonia is severe within the herd and represents an unfavorable health condition [23].

Regarding the scores of pleurisy lesions, there was a strong correlation between the severity of the lesions and the presence of *P. multocida* and *A. pleuropneumoniae*. In terms of lung samples, the severity of the lesions was correlated with the presence of *P. multocida* and *M. hyopneumoniae*. De Conti et al. [44] found a prevalence of 43.3% of *P. multocida* type A (65/150) in lungs without the occurrence of pleuritis in commercial farms in Brazil, which is probably related to the correlation between *M.a hyopneumoniae* and *P. multocida* type A infections [48]. They also found an average prevalence of 79.3% of *M. hyopneumoniae* positive samples by qPCR in the lung samples. The pathogenic mechanisms of this correlation are suggested to be due to the L-fucose composition increased by *M. hyopneumoniae*, which enhances the adherence of *P. multocida* to the bronchial and bronchiolar epithelial cells [49].

The adhesion of *A. pleuropneumoniae* to respiratory epithelial cells involves the binding of bacterial lipopolysaccharides to glycosphingolipids on the cell surface [50]. This process leads to biofilm formation, the liberation of toxins, and the development of lesions [51,52]. Our findings revealed a mean pleuritis observation of 11.47% and the presence of *A. pleuropneumoniae* in lung samples from all evaluated lots. A similar study [52] conducted in Europe reported a prevalence of pleuritis ranging from 12 and 41% in commercial farms of pigs with the presence of this pathogen. Furthermore, other bacteriological studies were limited in identifying the presence of *A. pleuropneumoniae* in chronic lesions since this pathogen is a known initiator of chronic pleuritis, which may be explained by the presence of *M. hyopneumoniae* and *P. multocida* in carcasses at slaughter [44].

In some cases, pigs that recover from acute diseases remain chronically infected without showing any clinical signs. However, these animals may develop chronic lung alterations, such as fibroblastic pleurisy and lung tissue sequesters surrounded by fibrotic tissue [12,53]. It is important to note that not all chronic pulmonary lesions are caused by bacterial pathogens; viral infections or environmental factors can also contribute to the development of these lesions. Moreover, different bacterial pathogens may have varying tendencies to spread from the lungs to the chest cavity, which may affect the specific sampling approach used depending on the suspected underlying pathogen. Our findings suggest a strong association between *A. pleuropneumoniae* and lesions characteristic of pleuropneumonia with a minor involvement of the pleura, similar to the study conducted by Ruggeri et al. [25]. In their study, *A. pleuropneumoniae* was largely associated with respiratory disease on the farm, as observed by clinical signs, but was not detected in fattening pigs at slaughter.

Regarding the presence of pathogens in the thoracic cavity lesions, it was observed that all pathogens were present in all scores of lung lesions. For the pleural samples, *P. multocida* was only observed in lesion scores 3 and 4, while the other bacterial pathogens were present in all lesion scores. Naturally, *P. multocida* type A alone typically does not cause disease in pigs [54]. However, it has been reported that highly pathogenic strains of *P. multocida* may be the primary agents causing pneumonia and septicemia in pigs [55,56,57], particularly when the strain carrying the *pfhA* gene [56]. The presence of this gene is associated with the occurrence of polyserositis in growing–finishing animals, as reported by Piva et al. (2023) [58]. Additionally, infection with *M. hyopneumoniae* and *P. multocida* can result in disease characterized by elevated rectal temperatures, a severe cough, and more extensive lung lesions [1,54]. These lung lesions, in direct contact with the thoracic cavity, may lead to inflammation of visceral pleura, which can cause bacterial colonization in the cavity fluids, leading to pleurisy lesions. Our findings strongly support a correlation between quantification of *M. hyopneumoniae* and *P. multocida* in severe lesions of SPES method.

Additionally, a large percentage of animals tested positive for *M. hyopneumoniae* in lung samples across all scores of lesions. In the pleura, an increase in the percentage of positive samples was observed in scores 3 and 4, which is consistent with the findings of Turni, et al. [59], who suggested that the prevalence of the pathogen increases with the severity of pleuritis scores. *Mycoplasma hyopneumoniae* alone causes bronchopneumonia, but when there are coinfections with primary pathogens such as *A. pleuropneumoniae* or secondary pathogens such as *P. multocida*, suppurative bronchopneumonia occurs, which is associated with pleurisy [60]. In a recent study conducted in Brazil, pleurisy was found to occur in 75.65% of vaccinated pigs at slaughter. Moreover, there was a significant association between body weight and pleurisy score 2 (*p* = 0.03), with high quantification of *A. pleuropneumoniae* and *P. multocida* in lung tissues detected by qPCR (C. S. Malcher, F. A. M. Petri, G. A. Aguiar, L. P. Arruda, G. Y. Storino, L. T. Toledo, K. Sonalio, F. Hirose and L. G. Oliveira, unpublished data), these findings highlight the real zootechnical and financial impact of these pathogens on swine production and emphasize the importance of pleuritis scoring and CVPC coupled with molecular diagnostics.

The most common pathogens in lung and pleura samples were *G. parasuis* and *S. suis*, and the prevalence of *S. suis* was also observed in studies conducted in Canada and Australia [59,61]. Ruggeri et al. [25] reported that *S. suis* is associated with pleural and pericardial lesions, mainly in post-weaning pigs. Fibrinous or fibrinopurulent pleuritis, peritonitis, or polyserositis were identified in pigs infected with both *S. suis* and *G. parasuis.* While *M. hyopneumoniae* and *P. multocida* may contribute to observed lesions and pleuritis scores, *G. parasuis* and *S. suis* may not have a significant impact in this context. It is important to note that the absence of a significant difference between the quantification of *G. parasuis* and *S. suis* and the observed lesions and pleuritis scores does not necessarily mean that these pathogens are not present or playing a role in the disease process.

Lastly, *S. suis* was found to be the most commonly detected pathogen in the lungs by qPCR, with a prevalence of 38.5% in pleurisy findings. However, there was no significant correlation between the quantification of *G. parasuis* and *S. suis* and the occurrence of lesions in lungs and carcass. We found a strong correlation between *M. hyopneumoniae* and *S. suis* in lung lesions of carcasses with a pleurisy score of 4. The capacity of *M. hyopneumoniae* to produce substrate for another microorganism due to ciliostasis induction and alteration of mucociliary tract [62] resulting in colonization of other pathogens such as *S. suis*. Other factors such as host immune response, co-infections with other pathogens, and environmental factors can also influence the development and severity of disease.

Post-mortem monitoring during slaughter is important for early detection of PRDC. This allows for affected animals to be identified and sampled, helping to prevent the spread of disease in the production chain. Additionally, health visits to farms allow veterinarians and animal health technicians to monitor animal health, identify clinical signs of respiratory diseases, and provide guidance to producers on prevention and control measures. Following vaccination protocols is also important to protect pigs against respiratory diseases such as Porcine Enzootic Pneumonia (PEP), swIAV and other respiratory infections. Moreover, respiratory disease diagnosis in swine is performed through clinical examinations, laboratory tests, and other diagnostic techniques. Early identification of respiratory diseases is essential to control the spread of the disease and minimize its impact on swine production.

## 5. Conclusions

This study identified five bacterial pathogens associated with PRDC, namely *M. hyopneumoniae, A. pleuropneumoniae*, *P. multocida*, *G. parasuis* and *S. suis*, and evaluated their correlation with the severity of lesions in pleural and lung samples. The results suggest that qPCR could be a useful tool for identifying and quantifying the causative agents of PRDC. When combined with other practices such as health surveillance on farms and improvements in vaccination programs, it could provide a comprehensive evaluation of the disease’s relevance, strength, and potential application as a surveillance tool for veterinarians in the field. Such an approach could contribute to the effective control and management of PRDC, thereby improving pig health and welfare, and reducing economic losses.

## Figures and Tables

**Figure 1 animals-13-01493-f001:**
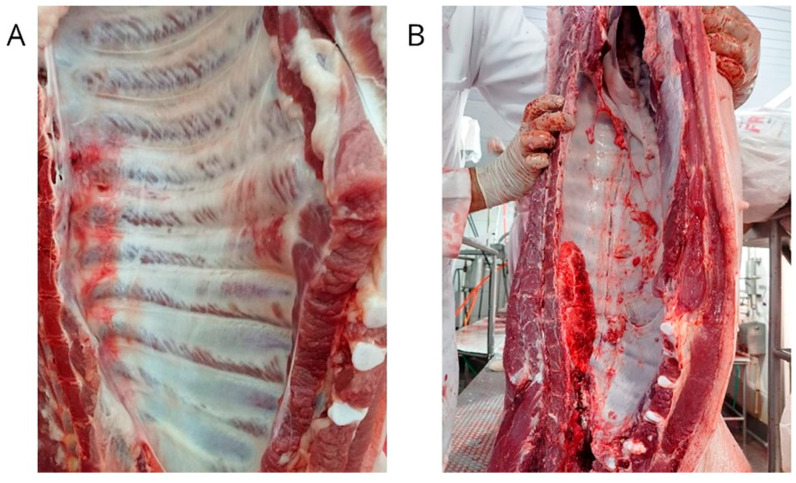
Observation of severe pleurisy lesions in pig carcasses. (**A**): pleurisy lesion scores 3; presence of marked inflammation in the chest cavity. (**B**): pleurisy lesion scores 4; evident presence of fibrin and adherence of lung lobes to the chest cavity of carcass.

**Table 1 animals-13-01493-t001:** Scoring of pleurisy lesions by the slaughterhouse pleurisy evaluation system (SPES) method.

Score	Features of Pleurisy, Considering the Extension and Localization of Lesions.
0	Absence of macroscopic lesions.
1	Pleurisy affecting the cranial-ventral portion of the lung; interlobar adhesion.
2	Discrete, unilateral pleurisy of the diaphragmatic lobe.
3	Discrete, bilateral pleurisy of both the diaphragmatic lobes; large, unilateral adhesion affecting the diaphragmatic lobe.
4	Large, bilateral adhesions between both the diaphragmatic lobes on one side and the chest cavity on the other.

**Table 2 animals-13-01493-t002:** Sequences of the primers and hydrolysis probes used for each target region of the multiplex qPCR–*omIA* gene (*A. pleuropneumoniae*), *p102* gene (*M. hyopneumoniae*), and *kmt1* gene (*P. multocida*), and the respective amplifier sizes.

Gene	N° of Access on GenBank	Primers and Probes	The Nucleotide Sequence (5′ → 3′)	Amplicon Size (bp)	Reference
*omIA*	[gb|NC_009053|]	F_App_ omIA	AGTGCTTACCGCATGTAGTGGC	153	[41]
		R_App_ omIA	TTGGTGCGGACATATCAACCTTA		
		P_App_ omIA	5′-Cy5-CGATGAACCCGATGAGCCGCC-3′-IB^®^RQ		
*p102*	[gb|AE017332.1|]	F_Mhp_p102	5′-TAAGGGTCAAAGTCAAAGTC-3′	150	[42]
		R_Mhp_p102	5′-AAATTAAAAGCTGTTCAAATGC-3′		
		P_Mhp_p102	5′-FAM-AACCAGTTTCCACTTCATCGCC-3′-BHQ1		
*kmt1*	[gb|NC_017027.1|]	F_Pmt_kmt1	5′-GGGCTTGTCGGTAGTCTTT-3′	148	[43]
		R_Pmt_kmt1	5′-CGGCAAATAACAATAAGCTGAGTA-3′		
		P_Pmt_kmt1	5′-TexRd- CGGCGCAACTGATTGGACGTTATT3′-IB^®^RQ		

*omIA:* target gene for *Actinobacillus pleuropneumoniae; p102:* target gene for *Mycoplasma hyopneumoniae; kmt1:* target gene for *Pasteurella multocida* type A; F: forward primer; R: reverse primer; P: probe; bp: base pairs.

**Table 3 animals-13-01493-t003:** Results of lung consolidation assessments, PI, and percentage of animals with pleurisy per herd.

	Lot 1	Lot 2	Lot 3	Lot 4	Lot 5	Lot 6	Mean	Median Standard Error	*p*-Value
Evaluated carcasses	128	151	107	111	113	116	121	14.91	-
Pulmonary consolidation (%)	9.05 ^a^	5.95 ^a^	10.00 ^a^	16.32 ^b^	14.57 ^b^	8.32 ^a^	10.70	3.61	*p* = 0.03
Pneumonia index (PI)	1.25 ^a^	1.03 ^b^	1.40 ^a^	1.89 ^a^	1.81 ^a^	1.35 ^a^	1.46	0.30	*p* = 0.04
Occurrence of pleuritis (%)	10.16 ^a^	5.95 ^a^	5.61 ^a^	18.92 ^b^	15.93 ^b^	17.24 ^b^	12.30	5.34	*p* = 0.03
App index	0.10 ^a^	0.00 ^b^	0.13 ^a^	0.21 ^a^	0.19 ^a^	0.17 ^a^	0.13	0.07	*p* = 0.04

^a,b^ For each lot, means followed by the same letter are not significantly different by using Tukey’s parametric test (*p* < 0.05).

**Table 4 animals-13-01493-t004:** The number of positive lung and pleura samples for *A. pleuropneumoniae*, *M. hyopneumoniae*, *P. multocida*, *G. parasuis*, and *S. suis*.

Groups	*A. pleuropneumoniae*		*M. hyopneumoniae*		*P. multocida*		*G. parasuis*		*S. suis*	
	Lungs	Pleura	Lungs	Pleura	Lungs	Pleura	Lungs	Pleura	Lungs	Pleura
G1 (*n* = 10)	4	2	10	1	3	0	9	10	9	9
G2 (*n* = 11)	2	6	10	1	5	0	11	11	11	9
G3 (*n* = 10)	8	5	9	5	4	2	9	10	8	6
G4 (*n* = 11)	5	4	11	4	8	4	9	11	11	8
G5 (*n* = 11)	5	4	10	2	2	0	10	10	11	10

**Table 5 animals-13-01493-t005:** Median of samples for each of the five pathogens in each score of pleurisy, of the lung and pleura samples. Medians followed by equal letters indicate that values do not differ from each other by using Dunn’s test at a significance level of *p* < 0.05.

	Lung Samples			Pleura Samples		
	Score of Pleurisy	Median *	Median Standard Error	Score of Pleurisy	Median *	Median Standard Error
*M. hyopneumoniae*	0	3.30 × 10^−1 b^	8.77 × 10^−1^	0	4.73 × 10^−1 a^	1.25 × 10^0^
	1	9.71 × 10^2 ab^	2.46 × 10^1^	1	6.40 × 10^−2 a^	1.62 × 10^−1^
	2	6.75 × 10^3 b^	1.72 × 10^2^	2	3.06 × 10^2 a^	8.47 × 10^−2^
	3	2.11 × 10^3 ab^	5.34 × 10^2^	3	4.36 × 10^1 a^	1.10 × 10^0^
	4	1.13 × 10^4 a^	9.35 × 10^2^	4	6.48 × 10^1 a^	1.72 × 10^0^
*P. multocida*	0	4.64 × 10^3 b^	1.23 × 10^−2^	0	0.00 × 10^0 b^	0.00 × 10^0^
	1	4.91 × 10^3 b^	1.31 × 10^−2^	1	0.00 × 10^0 b^	0.00 × 10^0^
	2	7.4 × 10^0 ab^	1.84 × 10^1^	2	0.00 × 10^0 b^	0.00 × 10^0^
	3	1.3 × 10^2 ab^	9.30 × 10^−2^	3	5.17 × 10^3 ab^	1.31 × 10^−2^
	4	8.37 × 10^1 a^	2.22 × 10^0^	4	7.68 × 10^1 a^	2.04 × 10^0^
*A. pleuropneumoniae*	0	6.84 × 10^2 bc^	1.81 × 10^−1^	0	6.83 × 10^0 a^	6.88 × 10^−2^
	1	7.42 × 10^3 bc^	1.88 × 10^−2^	1	6.24 × 10^0 a^	1.16 × 10^−2^
	2	1.20 × 10^4 a^	4.68 × 10^0^	2	6.68 × 10^0 a^	7.47 × 10^−1^
	3	2.06 × 10^2 ab^	5.22 × 10^2^	3	7.43 × 10^0 a^	4.25 × 10^−1^
	4	6.43 × 10^3 c^	1.71 × 10^−2^	4	6.60 × 10^0 a^	1.93 × 10^−1^
*G. parasuis*	0	1.15 × 10^1 a^	3.06 × 10^−1^	0	1.76 × 10^1 a^	4.68 × 10^−1^
	1	2.70 × 10^1 a^	6.84 × 10^−1^	1	2.23 × 10^1 a^	5.63 × 10^−1^
	2	4.66 × 10^2 a^	1.18 × 10^−1^	2	2.57 × 10^1 a^	7.11 × 10^−1^
	3	4.65 × 10^1 a^	1.18 × 10^0^	3	1.53 × 10^0 a^	3.87 × 10^0^
	4	4.20 × 10^1 a^	1.11 × 10^1^	4	1.66 × 10^1 a^	4.40 × 10^−1^
*S. suis*	0	9.44 × 10^−2 a^	2.50 × 10^−1^	0	1.65 × 10^−1 a^	4.39 × 10^−1^
	1	5.80 × 10^−2 a^	1.47 × 10^−1^	1	1.03 × 10^−1 a^	2.61 × 10^−1^
	2	4.21 × 10^−2 a^	1.09 × 10^−1^	2	2.20 × 10^−1 a^	6.10 × 10^−1^
	3	1.86 × 10^0 a^	4.71 × 10^0^	3	9.94 × 10^−2 a^	2.51 × 10^−1^
	4	8.03 × 10^−2 a^	2.13 × 10^−1^	4	1.40 × 10^−1 a^	3.70 × 10^−1^

* Equal letters indicate no significant differences by the non-parametric Dunn’s test (*p* > 0.05).

**Table 6 animals-13-01493-t006:** Spearman correlations data (*p*- and rho-values) for analysis of severity of SPES scores: (1) the quantification of *A. pleuropneumoniae* and *M. hyopneumoniae* were correlated with the macroscopic lesions of pleurisy score 2; (2) the quantifications of *G. parasuis* and *S. suis* were correlated with the macroscopic lesions of pleurisy score 3; and (3) the quantifications of *M. hyopneumoniae* and *P. multocida* and *M. hyopneumoniae* and *S. suis* were correlated with the macroscopic lesions of pleurisy score 4.

Pleurisy Lesion Severity	Interaction	Spearman’s Correlation	*p*-Value
Score 2	*A. pleuropneumoniae* and *M. hyopneumoniae*	R = 0.42	*p* = 0.04
Score 3	*G. parasuis* and *S. suis*	R = 0.45	*p* = 0.04
Score 4	*M. hyopneumoniae* and *P. multocida*	R = 0.58	*p* = 0.03
	*M. hyopneumoniae* and *S. suis*	R = 0.63	*p* = 0.02

## Data Availability

Not applicable.

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
