# Peer review of "Associations between Pleurisy and the Main Bacterial Pathogens of the Porcine Respiratory Diseases Complex (PRDC)"

_animals, 2023, doi:10.3390/ani13091493_

Round 1

Reviewer 1 Report

Comments for the authors

Major comments

Simple summary 

-Mycoplasma hyopneumoniae (M. hyopneumoniae), Actinobacillus pleuropneumoniae (A. pleuropneumoniae), Pasteurella multocida (P. multocida), Glaesserella parasuis (G. parasuis), and Streptococcus suis (S. suis)

- quantitative polymerase chain reaction (qPCR)

- you should clearly describe the aim of this study 

Abstract 

-        - you should clearly describe the aim of this study 

-         

Introduction

-Actinobacillus pleuropneumoniae (A. pleuropneumoniae), Bordetella bronchiseptica (B. bronchiseptica), Glaesserella parasuis (G. parasuis), Mycoplasma hyopneumoniae (M. hyopneumoniae), Mycoplasma hyorhinis (M. hyorhinis), Pasteurella multocida (P. multocida), and Streptococcus suis (S. suis). You should these abbreviated terms in all text.

- use ADWG or ADG in all text.

- declare one time ‘’quantitative polymerase chain reaction (qPCR)’’ and then use the term ‘’qPCR’’

Materials and Methods

-        provide details about the selection criteria for examined slaughtered pigs. 

-        which was the mean BW and age of animals?

-        More information about the vaccination program of the 8 commercial farms

-        Were vaccinated against respiratory diseases the examined animals (e.g. A. pleuropneumoniae, G. parasuis, M. hyopneumoniae)?

Results

-        Table 3: provide p-value and sd

Discussion:

-         you could discuss the economic impact of your results on Brazilian pig production.  

Minor comments

Abstract 

-        -… diseases affecting pigs include PRDC. In pigs, the infection may cause lesions such as pleurisy which..

Introduction 

-        at the slaughterhouse. The most common lesions are pneumonia and pleurisy [6]In the Brazilian pig industry

-        -… and environmental factors can also influence the development and severity of the disease.

-        - .. classified by the SPES method, in both pleura and lung…

Results 

-        The test showed significant positive correlations between the DNA loads of M. hyopneumoniae and P. multocidain qPCR, gross and macroscopic lung lesions and scores of pleuritis in carcasses. However, the correlation was moderate (Spearman's r-coefficient < 0.5) due to the wide distribution of pathogens quantification in each class of SPES scores.

Discussion

- In this study, our findings showed that A. pleuropneumoniae was strongly associated with lesions characteristic of pleuropneumonia with minor involvement of pleura, similar to a study conducted

Conclusions

-        .. as maintainers of pleuritis could be identified

-        .. as known causes of porcine pleuropneumonia,…

-        .. is alarming and a similar prevalence has been found..

Reviewer 2 Report

Major points

The study was performed very accurately and used high-quality laboratory and statistical methods. What is missing is a (are) concise conclusion(s), which limitation is also reflected by the very non-specific short summary. Thus, the reader is left alone with a lot of data and correlations, but it is not entirely clear, what the novel findings are. This means that the reader should be more precisely directed towards some probably unexpected correlations or non-correlations and their significance for basic and/or applied infectiology.

Additional points

Intense English editing should be performed, as multiple and severe linguistic mistakes are present throughout the manuscript. Only some examples: Simple summary: "pleurisy score were evaluated", "microscopically lesions", sometimes, even no complete sentences are given, for example first sentence after Table 5.

Simple summary: the last, non-specific sentence should be modified to highlight the true strength of the study, which are the sometimes surprising correlations or non-correlations between specific pathogen loads and pathological findings.

In the M&M section, 1015 animals from 6 different herds are reported, but from 2 different herds in the first line of the 3.1. section.

3.2. section: the first sentence should be deleted, as it is an instruction by MDPI. Also, how can reaction efficiencies be greater than 100 % (here 102.8 %)?

Table 5: what does the “sd” in brackets after the “Mean” indicate? Usually it is the abbreviation for “standard deviation”, but this is not given.

In the results section, the use of the Chi2-test is described, but this is not mentioned in the statistics section.

Table 6: contrary to the statement in the legend, no p values are given, but mean values for DNA loads. Additionally, it is unclear to what pathogen in each of the combinations (first or second) the given numbers refer to, or do the respective numbers account for both pathogens of each combination. Please clarify.

Discussion: Paragraph dealing with PI: “… while a value of less than 1 is considered severe….”. Shouldn't it be "more than..."?

First sentence of the last paragraph: “Moreover, this pathogen...” Which pathogen is meant? S. suis or G. parasuis?

Round 2

Reviewer 1 Report

No comments 

Reviewer 2 Report

Most of the points are now met satisfactorily. Only the first sentence of the "Conclusions" section is incomplete as you forgot to mention the specific pathogens causing PRDC and wrote "..." instead.